# Coding Metamaterial Analysis Based on 1-Bit Conventional and Cuboid Design Structures for Microwave Applications

**DOI:** 10.3390/ma15217447

**Published:** 2022-10-24

**Authors:** Tayaallen Ramachandran, Mohammad Rashed Iqbal Faruque, Mohammad Tariqul Islam, Mayeen Uddin Khandaker, Hamid Osman, Imed Boukhris

**Affiliations:** 1Space Science Centre (ANGKASA), Institute of Climate Change (IPI), Universiti Kebangsaan Malaysia, Bangi 43600, Malaysia; 2Department of Electrical, Electronic & Systems Engineering, Universiti Kebangsaan Malaysia, Bangi 43600, Malaysia; 3Centre for Applied Physics and Radiation Technologies, School of Engineering and Technology, Sunway University, Bandar Sunway 47500, Malaysia; 4Department of General Educational Development, Faculty of Science & Information Technology, Daffodil International University, DIU Road, Dhaka 1341, Bangladesh; 5Department of Radiological Sciences, College of Applied Medical Sciences, Taif University, Taif 21944, Saudi Arabia; 6Department of Physics, Faculty of Science, King Khalid University, P.O. Box 9004, Abha 62529, Saudi Arabia; 7Laboratoire des Matériaux Composites cÉramiques et Polymères (LaMaCoP), Département de Physique, Faculté des Sciences de Sfax BP 805, Université de Sfax, Sfax 3000, Tunisia

**Keywords:** 1-bit coding metamaterial, cuboid coding metamaterial, FR-4, phase response, RCS reduction

## Abstract

This study aimed to investigate the compact 1-bit coding metamaterial design with various conventional and cuboid shapes by analysing the bistatic scattering patterns as well as the monostatic radar cross-section for microwave applications. The construction of this metamaterial design depends on binary elements. For example, 1-bit coding metamaterial comprises two kinds of unit cell to mimic both coding particles such as ‘0’ and ‘1’ with 0° and 180° phase responses. This study adopted a 1 mm × 1 mm of epoxy resin fibre (FR-4) substrate material, which possesses a dielectric constant of 4.3 and tangent loss of 0.025, to construct both elements for the 1-bit coding metamaterial. All simulations were performed using the well-known Computer Simulation Technology (CST) software. The elements were selected via a trial-and-error method based on the phase response properties of the designs. On the other hand, the phase response properties from CST software were validated through the comparison of the phase response properties of both elements with the analytical data from HFSS software. Clear closure was obtained from these findings, and it was concluded that the proposed conventional coding metamaterial manifested the lowest RCS values with an increasing number of lattices. However, the cuboid-shaped design with 20 lattices demonstrated an optimised bistatic scattering pattern of −8.49 dBm^2^. Additionally, the monostatic RCS values were successfully reduced within the 12 to 18 GHz frequency range with −30 to −10 dBm^2^ values. In short, the introduced designs were suitable for the proposed application field, and this unique phenomenon is described as the novelty of this study.

## 1. Introduction

The scientific world has recently grown accustomed to coding metamaterial. The incorporation of coding sequences in the metamaterial array structure can result in a coding metamaterial that manipulates electromagnetic (EM) waves to achieve various functionalities. A coding metamaterial can be constructed by simply applying various coding sequences once the elements have been selected based on the adopted bits. For instance, 1-bit coding metamaterial possesses two types of unit cell to mimic ‘0’ and ‘1’ binary elements that exhibit 0 and 180° phase responses [1]. The antenna and optical communities widely utilise this type of opposite-phase method. Meanwhile, the construction of a 1-bit coding metamaterial consists of a 0 phase element with a perfect magnetic conductor and a 180° phase element that is composed of perfect magnetic conductor cells with perfect electric conductor cells in a chessboard-like configuration. Therefore, this allows the reflections of any normally incident EM plane waves to cancel out, resulting in the reduction of RCS values. This type of metamaterial is dissimilar to the conventional metamaterials that are based on the macroscopic scale and which are commonly defined according to effective medium parameters and categorised as analogue metamaterials.

Conventional metamaterial designs are widely utilised in many applications to gain optimised performances, for instance, in specific absorption rate (SAR) or EM reduction applications, sensors, medical applications, optical filters, antenna applications, etc. [2,3,4,5,6,7,8,9,10]. In 2021, Ramachandran et al. investigated the practicability of a metamaterial for SAR reduction of 5G EM energy in human head tissue [11]. This study focused on building seven compact and novel square-shaped metamaterial structures on an FR-4 substrate material for SAR reduction applications. Meanwhile, the same first author introduced a multi-layered, square-shaped metamaterial (MSM) for EM reduction applications in 2022 as well [12]. The focus of their research was to decrease the SAR values for the sub-6 frequency range by designing a compact and multi-layered structure. Hence, this study adopted only six square-shaped metamaterial rings designed on a Rogers RO3006 substrate material with a thickness of 0.25 mm. On the other hand, Hossain et al. introduced a planar multiple-inputs multiple-outputs (MIMO) antenna consisting of two sets of orthogonally placed 1 × 12 linear antenna arrays for 5G millimetre wave applications in 2021 [13]. A 90 mm × 160 mm Rogers RT/Duroid 5880 grounded substrate material was adopted in this work. Moreover, Pandya et al. designed a multi-layered antenna for navigation, Wi-Fi, and satellite applications [14]. Wireless communications play an important role in transmitting information from point to point. User demands for wireless communications are met with compact-sized devices that are cost effective and have advanced intelligence. These literature reviews revealed that most of the studies focused on a compact-sized design to achieve the desired applications. The selection process of substrate materials is very important for the initial stage of coding design construction. Hence, the substrate material known as FR-4 was adopted because it is suitable for the construction of coding metamaterial design since it is a low-cost material and can be easily obtained on the market. 

Conventional metamaterial design is also commonly used in radar cross-section (RCS) reduction applications. An approach for wideband RCS reduction of a microstrip array antenna was presented and discussed in a previous study [15]. The scheme was based on the microstrip resonators and absorptive frequency selective surface (AFSS). A polarisation conversion metasurface (PCM) with a wide operating band and high polarisation conversion ratio was proposed for antenna RCS reduction by Hong et al. [16], and they constructed a third-order quadrate fractal structure. The PCM was applied for the RCS reduction application of microstrip antennas using a chessboard-like arrangement. The performance of the proposed PCM and antenna was assessed through measurement and simulation. In addition to this, Zhang et al. investigated the RCS reduction of a patch antenna by adopting a metamaterial absorber (MA) [17]. The introduced unit cell MA had a height and thickness of 5.5 mm and 0.3 mm, respectively. The study also compared the patch antenna patterns and reflection coefficients for loading and unloading the metamaterial. Recently, Liang et al. proposed and experimentally realised novel, diatomic metasurfaces for full-Stokes polarisation perfect absorption in the mid-infrared [18]. Overall, the chessboard-like configuration yielded promising outcomes for the RCS reduction application. Therefore, since coding metamaterial is generally applied for the RCS reduction application, this configuration is worth exploring.

On the other hand, coding metamaterial has also been investigated in several studies. In 2014, Cui et al. proposed two-step digital metamaterials and the construction of coding metamaterials by adopting 1 bit and introducing unique metamaterial particles that have either a ‘0’ or ‘1’ response controlled by a biased diode [19]. A 2-bit coding metamaterial was also analysed in this study. In the 2-bit coding, four types of unit cell, with phase responses of 0, π/2, π, and 3π/2, were utilised to mimic the ‘00’, ‘01’, ‘10’, and ‘11’ elements, respectively. They also proposed a unique metamaterial particle that has either a ‘0’ or ‘1’ response controlled by a biased diode. Based on this phenomenon, the author introduced digital metamaterials with unit cells that possess either a ‘0’ or ‘1’ state. Using a field-programmable gate array, the digital control over the digital metamaterial was realised. Programming of different coding sequences allows a single digital metamaterial to manipulate EM waves in different manners, thereby, realising programmable metamaterials. A transmission-type coding metasurface was suggested and experimentally proven by Liu et al. in 2016 [20] to bend normally occurring terahertz beams in abnormal directions and produce non-diffractive Bessel beams in both normal and oblique directions. Meanwhile, Moeini and Cui proposed fractal coding metamaterials which can be used to design reflective metasurfaces with self-similar, pseudo-random phase responses based on a coding strategy utilising fractal interpolation functions [21]. In contrast, several research works such as [22,23,24] investigated coding metamaterial for antenna and microwave applications using various designs in recent years.

Furthermore, several other unique research fields have been adopted by scholars, such as a hyperbolic metamaterial and metasurface for terahertz frequency. For instance, Danila [25] conducted a theoretical analysis of the spectroscopic characteristics of a straightforward, hybrid metasurface made up of a periodic array of an Si–Au pattern in the mid-infrared spectral range in 2020. Meanwhile, the same author [26] proposed theoretical investigations in the terahertz G band for a piezoelectrically responsive ring–cone element metasurface composed of polyvinylidene fluoride (PVDF)/silicon and PVDF/silica glass. On the other hand, Gric [27] investigated a different design, aiming to obtain effective, tuneable THz amplifiers with small dimensions and lasers with broadband operation based on active THz hyperbolic metamaterials. Meanwhile, coding metamaterial has high demand in technological development as it is simple, time saving, and compatible with digital devices. Although coding metamaterials have excellent properties, limited studies have assessed them to date because coding-based metamaterial is relatively new to the scientific community, thus, requiring extensive research to realise its various properties in a wide range of applications. In addition to this, the previously proposed coding metamaterial designs were generally bigger in size and only explored the coding patterns in lattice forms. Furthermore, the construction of unit cells for the desired number of bits is tricky because not all metamaterial designs exhibit the desired phase response properties. Hence, this study proposed the utilisation of compact unit cells with 1 mm × 1 mm dimensions to construct conventional and cuboid 1-bit coding metamaterial designs by adopting various coding sequences for microwave applications. 

## 2. Unit Cell Elements and Properties

All numerical simulation analyses were performed using Computer Simulation Technology (CST 2019) software. The frequency-domain solver and tetrahedral mesh were adopted to calculate the phase response property. Meanwhile, the RCS values were simulated by adopting a time-domain solver and hexahedral mesh for both conventional and cuboid coding metamaterial. The selection process of both unit cells to mimic ‘0’ and ‘1’ elements for the 1-bit coding metamaterial was performed via a trial-and-error method. Several constraint variables were set to yield an optimal design structure in the trial-and-error method. Section 3.1 explores one of the design parametric studies. The elements in the 1-bit coding metamaterial possessed two varied phase responses, namely 0 and π. Upon completing the unit cell selection process, the unit cell metamaterial design, illustrated in Figure 1a,b, was chosen to mimic element ‘1’, whereas a substrate material with a similar dimension and thickness to the design of element ‘1’ was selected to mimic element ‘0’. The two elements, however, were not dependent on macroscopic, medium factors. As a result, Figure 1c shows the phase responses and phase differences of the elements. Throughout the whole frequency range, from 13.25 GHz to 18 GHz, the chosen element ‘1’ successfully displayed a 180 phase difference response. However, the absolute phase response of the suggested element ‘0’ might not show zero response at a certain frequency. Since the phase can be normalised to zero, it does not have an impact on any aspects of physics. Even though the phase response characteristics were only visible between 13.25 GHz and 18 GHz, this study used a frequency range from 0 to 18 GHz to examine response modifications. On the other hand, the High-Frequency Structure Simulator (HFSS) programme was also used to validate the phase response properties. As seen in Figure 1c, the phase response characteristics from the HFSS programme produced results that were nearly identical to those from the CST 2019 software.

Only the substrate material without any metamaterial design was selected as element ‘0’. However, both the proposed elements were composed of a compact and similar substrate material known as FR-4 with a dimension of 1 mm × 1 mm. A 0.98 mm copper material was constructed on the substrate material for element ‘1’. Then, two sets of rectangular bars were vertically placed on both the left and right sides of the copper material to subtract them, as demonstrated in Figure 1a. Moreover, two sets of horizontal, rectangular bars were also placed in the middle of the design to subtract the copper material. Meanwhile, three circles were subtracted from the left and right sides of these horizontal bars. However, the adopted dimension of vertical bars designed earlier differed from these horizontal bars. Table 1 demonstrates the dimensions of the proposed element ‘1’. The selection process of the elements was performed through their phase responses by adopting a frequency-domain solver. Once both elements were selected, the constructed coding metamaterials were simulated using a time-domain solver to yield properties of monostatic RCS and bistatic scattering patterns.

## 3. Parametric Studies 

This study incorporated several parametric studies for performance analysis, including dimensions of various element ‘1’ metamaterial design analyses, several numbers of lattices for conventional metamaterial, and cuboid-shaped coding metamaterial. All simulations in CST analysed the phase responses, monostatic RCS, and bistatic scattering patterns. RCS is an object’s EM signature and assesses an object’s radar detectability. The RCS is mostly used in military technologies, such as when an aircraft infiltrates an enemy region undetected. Any anti-aircraft missile may quickly annihilate an aircraft that is detected. As a result, RCS is frequently used in military aircraft technology. As a result, it is a crucial area of research in the antenna and radar engineering fields. RCS is derived from several factors. A target’s RCS can be expressed by Equation (1) if it is located by the radar at a distance of R. Moreover, the RCS is primarily measured in terms of square metres or in a more conventional unit known as dBsm (dBm^2^).
σ = 4πR^2^ (P_r_/P_i_)(1)

σ = radar cross-section (m^2^ or dBm^2^);

R = distance between the target and radar receiver (m);

P_r_ = backscattered or reflected power density (Watt/m^2^);

P_i_ = incident power density on target (Watt/m^2^).

It is also known as the ratio of the backscattered power density (P_r_) to the incident power density (P_i_) on the target. The factors influencing the RCS values include the physical geometry of the material, the direction of radar, polarisation of the scattered signal, and signal frequency.

### 3.1. Analyses of Various Metamaterial Design Dimensions (c) for Element ‘1’

Once element ‘0’ was selected, as discussed in Section 2, the selection process of element ‘1’ was performed by adopting a trial-and-error method. Therefore, four different c values, 1 mm, 0.98 mm, 0.92 mm, and 0.88 mm, were adopted in this parametric study, and the performances were plotted in two graphs, as illustrated in Figure 2a,b. On the other hand, several parameter changes inside the metamaterial design were performed as well as changes to the c values. However, only changing the c value manifested the desired phase response properties that we were looking for. Moreover, changing the design structure, for example, the size of the circle, did not manifest a relative effect on the phase response property. The results revealed that the unit cell design with a c of 0.98 mm produced a 180° phase difference between the elements at the desired frequency. Overall, when c values were decreased, inconsistent phase responses were recorded. Although the changing of the c values manifested unique phase response properties when compared to other parameters, all four designs exhibited identical monostatic RCS values, as shown in Figure 2b. The proposed designs manifested RCS values ranging from −85 dBm^2^ to −75 dBm^2^ at a frequency range of 12 GHz to 18 GHz, respectively. Therefore, it can be concluded that the c values in this work directly influenced the phase response properties. Moreover, this unit cell design only generated a spherical scattering beam pattern, which was enabled by various coding sequences. Meanwhile, the redirection of EM waves along numerous paths was made possible by adopting a random arrangement of ‘0’ and ‘1’ coding particles in the desired number of lattices. 

### 3.2. Number of Lattices

Coding metamaterial can manipulate EM waves so a normally incident beam can reflect in more than one oriented direction under various coding sequences. To assess the mentioned physical phenomenon, a general, square-shaped metamaterial design composed of N × N equal-sized lattices was adopted by occupying the proposed elements. Therefore, this study adopted several lattices, 8, 12, 16, and 20, respectively, to determine the optimal design structure. Since the unit cell design assumed the dimension of 1 mm × 1 mm, the lattice dimensions were 8 mm × 8 mm, 12 mm × 12 mm, 16 mm × 16 mm, and 20 mm × 20 mm. For each lattice, both the monostatic and bistatic scattering patterns of the proposed coding metamaterial were investigated using three different coding sequences. Moreover, the minimal monostatic RCS and bistatic scattering pattern at 15 GHz of each lattice were adopted as the best coding sequence. 

#### 3.2.1. Eight Lattices

Figure 3a–c illustrates the various coding sequences adopted for the coding metamaterial design with eight lattices made by rearranging elements ‘0’ and ‘1’. The variation in the coding sequences influenced the monostatic RCS and bistatic scattering pattern behaviours, as depicted in Figure 3d. Based on observation, Coding Sequence 3 yielded the lowest RCS values, ranging from −40 dBm^2^ to −30 dBm^2^ at 12 GHz to 18 GHz. Meanwhile, the maximum bistatic scattering pattern values reached −35.2 dBm^2^ for the same sequence. Overall, the three coding sequences only exhibited slight discrepancies between them, and the RCS reduction was directly influenced by the arrangement of elements ‘0’ and ‘1’ in the lattice form. The scattering pattern also manifested a similar style for the three coding sequences.

#### 3.2.2. Twelve Lattices

Based on the observation of monostatic RCS and bistatic scattering patterns in this section, it can be seen that the increased number of lattices affected the RCS reduction values for the three different coding sequences. The coding sequences, as demonstrated in Figure 4a–c, indicated slightly better reduction performances for both monostatic RCS and bistatic scattering patterns compared to the previous lattice (shown in Figure 4d). Although the scattering patterns of these three sequences manifested equivalent behaviours to those of the previous parametric study, the reduction in RCS values was successfully recorded. This is because the coding metamaterial with bigger lattices had more freedom to control the EM wave (likely in 16 and 20 lattices) and the arrangement of coding particles and had a major effect on the scattering pattern behaviour. The RCS values that were recorded at 12 GHz to 18 GHz were between −38 dBm^2^ and −29 dBm^2^, whereas the eight lattices manifested from −40 dBm^2^ to −30 dBm^2^. Coding Sequence 1, where the maximum bistatic reduction scattering pattern was measured at −28.9 dBm^2^, exhibited the optimal result compared to the rest of the sequences in this lattice. This analysis proved that better RCS reduction values can be yielded when the number of lattices is increased. However, this leads to a major size constraint variable that has to be dealt with when either constructing a metamaterial structure or practically applying a slightly bigger coding metamaterial for industrial applications.

#### 3.2.3. Sixteen Lattices

Figure 5a–d illustrates the bistatic scattering patterns of the three different coding sequences and monostatic RCS values. As demonstrated in previous subsections, an increased number of lattices enables optimised RCS reduction values. The monostatic RCS values at 12 GHz to 18 GHz ranged from −33 dBm^2^ to 29 dBm^2^, while 12 lattices exhibited values in the range of −38 dBm^2^ to −29 dBm^2^. In this analysis, Coding Sequence 2 produced the lowest RCS value curve and successfully exhibited bistatic scattering patterns at −25.7 dBm^2^. Therefore, the arrangement of elements ‘0’ and ‘1’ is crucial in coding metamaterial to obtain unique properties, such as control of EM wave, as it shapes the near-field as well as the far-field distribution of microwave frequency by simply adopting these coding sequences. 

#### 3.2.4. Twenty Lattices

Lastly, the analysis of lattices in this study was concluded by exploring the biggest coding metamaterial dimension of 20 mm × 20 mm as demonstrated in Figure 6a–c. Comparatively, there was an approximate 30% reduction difference that successfully occurred between the minimum bistatic scattering pattern values at 15 GHz for 8 and 20 lattices. Based on the analysis as shown in Figure 6d, Coding Sequence 1 for 20 lattices yielded greater RCS reduction values, ranging from −28 dBm^2^ to −23 dBm^2^ in the frequency range of 12 to 18 GHz. Meanwhile, the 16 lattices only manifested an RCS reduction range from −33 dBm^2^ to 29 dBm^2^. Therefore, the RCS value can be reduced with larger lattices; however, the usability is limited in terms of practical application since the current technological development mostly requires miniaturised devices. Hence, a unique idea or concept is required to reduce RCS values by satisfying the size constraint variable. However, the proposed 20-lattices coding metamaterial in this section is categorised as minimal size since it has a compact unit cell design. Therefore, it can be used in a few application fields that typically need smaller metamaterial designs to optimise performance.

### 3.3. Cuboid-Shaped Coding Metamaterial

This study proposed a cuboid-shaped coding metamaterial design to further reduce the RCS values. This subsection is divided into four categories in which optimised lattice designs were adopted to construct three different cuboid shapes. The monostatic RCS and bistatic scattering patterns of the proposed cuboid coding metamaterial were also investigated. The initial analysis indicated promising RCS reduction values compared to conventional coding metamaterial. Since the existing literature in the coding metamaterial field only focused on the conventional lattices of coding design structure, this study introduced a unique structure by adopting 1-bit coding elements. 

#### 3.3.1. Eight Lattices

Coding Sequence 3 in the eight lattices of conventional coding metamaterial analysis was adopted to construct three different cuboid shapes, as illustrated in Figure 7a–c. As predicted, the cuboid-shaped coding metamaterial exhibited a further reduction in RCS values compared to the conventional coding metamaterial. The proposed cuboid shapes yielded RCS values ranging from −43 dBm^2^ to −25 dBm^2^, as denoted in Figure 7d. Meanwhile, the maximum bistatic scattering patterns (−23.5 dBm^2^) were yielded by adopting Cuboid Shape 2. Cuboid Shape 3 exhibited a slightly higher bistatic value (−23.8 dBm^2^) compared to the previous shape.

#### 3.3.2. Twelve Lattices

For this set of analyses, Coding Sequence 1 was selected to construct the cuboid coding metamaterial design, as demonstrated in Figure 8a–c. At a frequency range of 12 GHz to 18 GHz, the proposed cuboid shapes exhibited monostatic RCS values ranging from −42 dBm^2^ to −23 dBm^2^ (as shown in Figure 8d), whereas the eight lattices gained a reduction range from −43 dBm^2^ to −25 dBm^2^. In addition to this, Cuboid Shapes 1 and 3 recorded the highest bistatic scattering patterns, −27 dBm^2^ and −15.8 dBm^2^, respectively. Furthermore, for the eight lattices, Cuboid Shapes 2 and 3 (designs with the lowest RCS values) exhibited bistatic values below −25 dBm^2^ compared to the 12 lattices, which produced RCS values below −20 dBm^2^ for Cuboid Shape 3. In short, a bigger cuboid shape yields slightly better required outcomes.

#### 3.3.3. Sixteen Lattices

In the 16 lattices, Coding Sequence 2 with the bistatic value of 25.7 dBm^2^ was adopted to construct the cuboid shapes, as illustrated in Figure 9a–c. All three cuboid shapes manifested monostatic RCS values ranging from −47 dBm^2^ to −17 dBm^2^, while the range from −42 dBm^2^ to −23 dBm^2^ was recorded in 12 lattices. Moreover, the proposed cuboid shapes produced bistatic values of −15.7 dBm^2^, −9.84 dBm^2^, and −22.8 dBm^2^, respectively. On the whole, the bistatic values decreased initially before increasing with increased cuboid size, as illustrated in Figure 9c.

#### 3.3.4. Twenty Lattices

The optimised monostatic RCS and bistatic scattering patterns were successfully manifested in the 20 lattices of the cuboid-shaped coding metamaterial designs. Coding Sequence 2, with a bistatic value of −23.5 dBm^2^, was adopted to construct the cuboid shapes demonstrated in Figure 10a–c. The monostatic RCS values that were recorded ranged from −35 dBm^2^ to −14 dBm^2^ (as shown in Figure 10d) and were better relative to the cuboid shapes of 16 lattices, which manifested with a range from −47 dBm^2^ to −17 dBm^2^. The cuboid shapes yielded maximum bistatic scattering pattern values of −16.5 dBm^2^, −16.4 dBm^2^, and −8.49 dBm^2^, respectively. In addition to this, the direction of polarisation of Cuboid Shape 3 is illustrated in Figure 10e. Overall, the cuboid-shaped coding metamaterial analysis demonstrated a further reduction in RCS values, whereby the optimisation of RCS values depended on the arrangement of elements and the number of cuboid shapes used.

### 3.4. Comparison between Lattices and Cuboid-Shaped Coding Metamaterial

The creation of coding metamaterial depends on the most crucial qualities, known as the digital characterisation, of every unit cell designed by binary codes, as stated in Section 2. This work used a 1-bit metamaterial for coding applications with two different types of unit cell that were suggested to represent the ‘0’ and ‘1’ elements, which have opposite phase reflections of 0° and 180°. For the analysis of the cuboid-shaped coding metamaterial, the same elements were used. Certain functions can appear when these coding particles are arranged in specific patterns on a two-dimensional plane. Analytical investigations were carried out in this study to precisely predict the scattering pattern of the suggested coding particles in N × N equal-sized lattices. In addition to the typical chessboard-like layouts, a few unique coding metamaterial design configurations were also used in this work. These configurations were built with the 0 phase, which resulted in a perfect magnetic conductor, and with the 180 phase, which had both perfect magnetic and perfect electric conductors. Therefore, any ordinarily encountered EM wave’s reflections would typically cancel out, resulting in a decrease in RCS values. For instance, the 16 lattice designs produced the first coding metamaterials with fundamentally distinct capabilities and intriguing coding sequences.

Figure 5d shows how a regularly incident plane wave generated by the 16 lattices of Coding Sequences 1 and 2 was diverted from a spherical shape to numerous directions. Additionally, as opposed to the spherical shape responses in lower lattices, coding patterns in the next number of lattices showed several scattering pattern directions. For 1-bit coding metamaterial designs, the radiation pattern is often symmetric concerning the x–z or y–z axes [1]. As a result, 1-bit coding metamaterial in lattice form makes it challenging to realise asymmetrical radiation patterns. Therefore, to obtain a distinct scattering beam direction, researchers should choose 2-bit or higher-bit design structures. However, the cuboid-shaped metamaterial designs that were suggested in this study were able to exhibit asymmetrical radiation patterns that indicated an anomalous direction brought on by the gradient phase distribution. Due to the 2-bit or higher number of bits possessing multiple elements with distinct phase response properties, the proposed design has more freedom to control the EM waves. Moreover, several works of literature proposed previously (refer to Table 2) have slightly larger coding elements, and it is a real challenge when designing cuboid shapes to satisfy the size constraint variable. Therefore, the authors wanted to incorporate this unique property in 1-bit coding metamaterial by adopting the smallest design structure. The conventional coding metamaterial structure in lattice form only has limited control over EW waves. Hence, as well as single, compact cuboid-shaped coding metamaterial, unique combinations of these cuboid shapes which have closed-environment design structures were proposed in this work to gain more control over EM waves. This was proved by the indication that the cuboid-shaped coding metamaterial possessed control over EM waves to manifest asymmetrical scattering patterns. Thus, the uniqueness of this study was defined as this distinct phenomenon caused by adopting a compact design. In addition, one of the primary surprises in this study was the size of the suggested elements. This work concentrated on miniaturised unit cell design because earlier literature suggested larger-sized unit cell designs and lattices. Additionally, the cuboid-shaped coding metamaterial designs demonstrated superior monostatic RCS reduction in comparison to the traditional coding metamaterial designs based on the data collected.

Moreover, the comparison of previously published, related works was compared with the proposed design, as tabulated in Table 2. All the references in this table either adopted many types of metamaterial, often hybrid metamaterial absorbers, polarisation conversion metamaterial, and coding metamaterial, or analysed the monostatic RCS values. Meanwhile, the literature reviews proposed various-sized designs and exhibited distinct results. For example, the biggest metamaterial design structure in [31] successfully produced near-zero monostatic RCS values at the absorption frequency range from 8.2 to 9.2 GHz. Meanwhile, the second largest metamaterial designs, in [28,30], with dimensions of 80 mm × 80 mm, achieved −30.1 dBm^2^ and −23 dBm^2^ at the peak point and 4.3 GHz, respectively. On the other hand, by adopting coding-based metamaterial, the monostatic RCS values can be reduced by adopting a smaller design structure, as in [29]. Meanwhile, the proposed work successfully introduced the smallest coding metamaterial design and explored the novel, unique, cuboid metamaterial structures for RCS reduction application.

## 4. Conclusions

This study described the performance changes due to the integration of several cuboid-shaped, 1-bit coding metamaterials for microwave frequency applications. The proposed 1-bit coding metamaterial designs were analysed based on the monostatic RCS and bistatic scattering pattern properties. The selection process of the ‘0’ and ‘1’ elements was performed via a trial-and-error method, and the phase responses were also investigated for each element. Although all numerical simulations were conducted using CST software, the phase response properties were validated using HFSS software. The numerical simulations demonstrated that the 20 mm × 20 mm coding metamaterial design in conventional lattices exhibited the highest bistatic scattering pattern of −23.5 dBm^2^. Moreover, the lowest monostatic RCS values were recorded for the 20-lattices coding metamaterial at frequencies ranging from 12 GHz to 18 GHz, corresponding to −30 dBm^2^ to −20 dBm^2^ RCS values. Moreover, the analyses revealed that the cuboid-shaped coding metamaterials reduced the RCS values compared to the coding metamaterials in distinct lattice shapes. When the size of the cuboid designs was increased, a maximum reduction in RCS values was observed. As such, Coding Shape 2 for 16 lattices manifested the maximum bistatic scattering pattern value of −9.84 dBm^2^. However, the rest of the lattices had the lowest bistatic scattering peak values when the size of the cuboid designs was increased. Meanwhile, Cuboid Shape 3 for 20 lattices demonstrated the highest bistatic scattering pattern value of 8.49 dBm^2^. In a nutshell, the RCS values were successfully reduced at microwave frequencies by adopting a compact and novel, 1-bit coding metamaterial design. In addition to this, the introduction of the cuboid-shaped coding metamaterial further reduced the RCS values compared to the conventional coding metamaterial design. Therefore, both conventional and cuboid-shaped coding metamaterial can be applied in a wide range of applications such as radar detection, stealth technology, military vehicles, etc.

## Figures and Tables

**Figure 1 materials-15-07447-f001:**
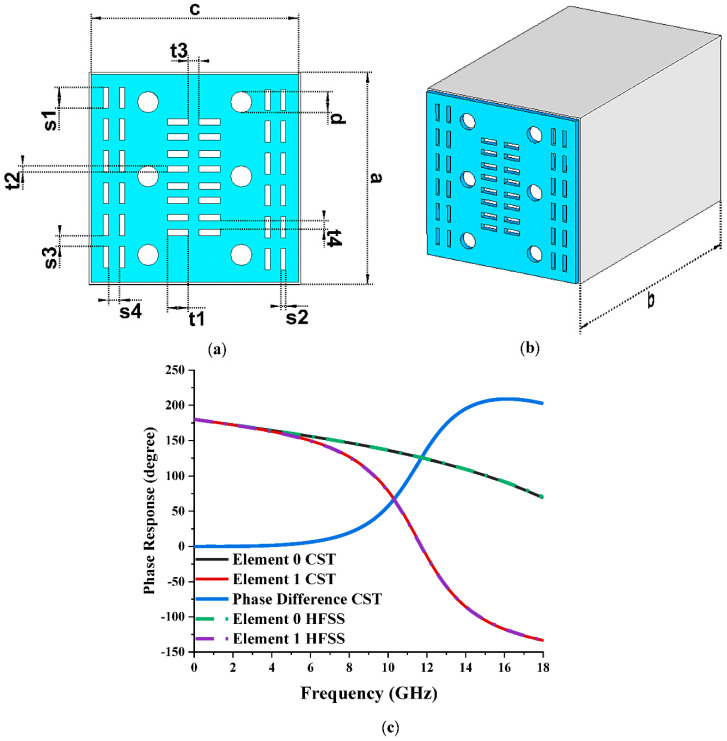
Numerical simulation geometry: (**a**) front view; (**b**) side view; (**c**) phase responses of elements ‘0’ and ‘1’.

**Figure 2 materials-15-07447-f002:**
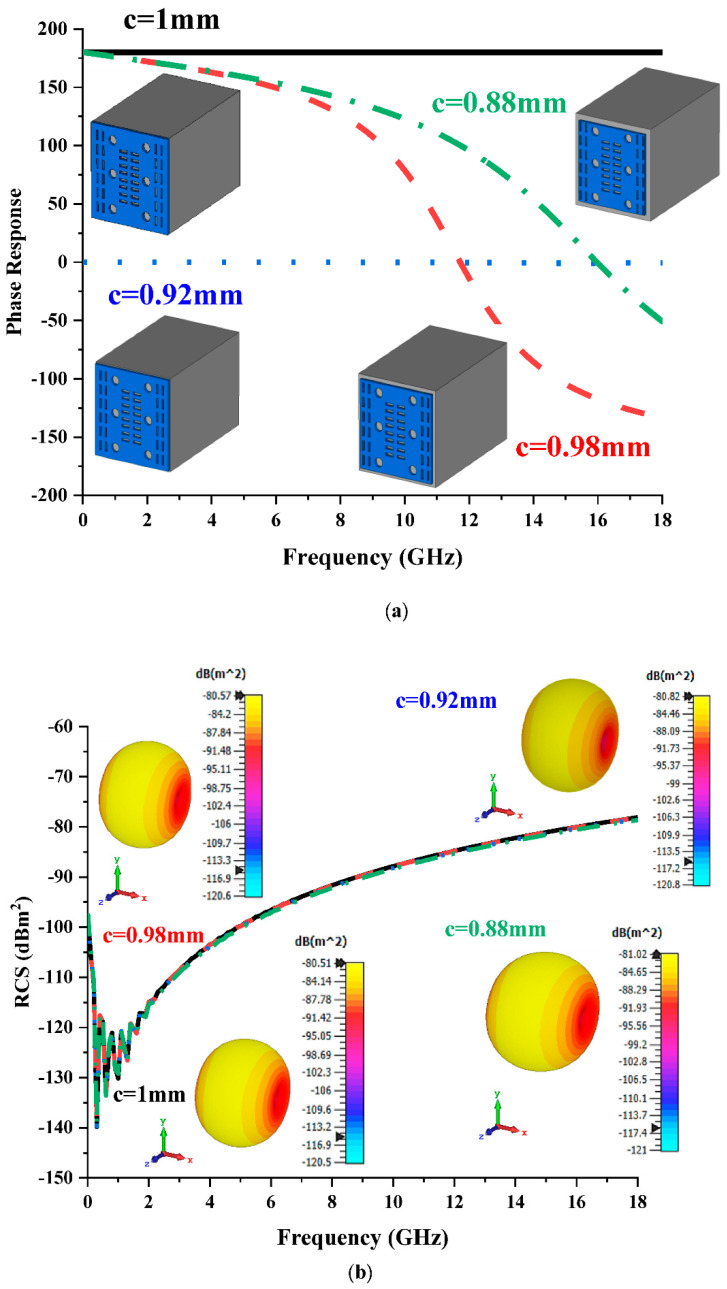
Element ‘1’ with four different c values: (**a**) phase responses; (**b**) monostatic RCS and bistatic scattering pattern at 15 GHz.

**Figure 3 materials-15-07447-f003:**
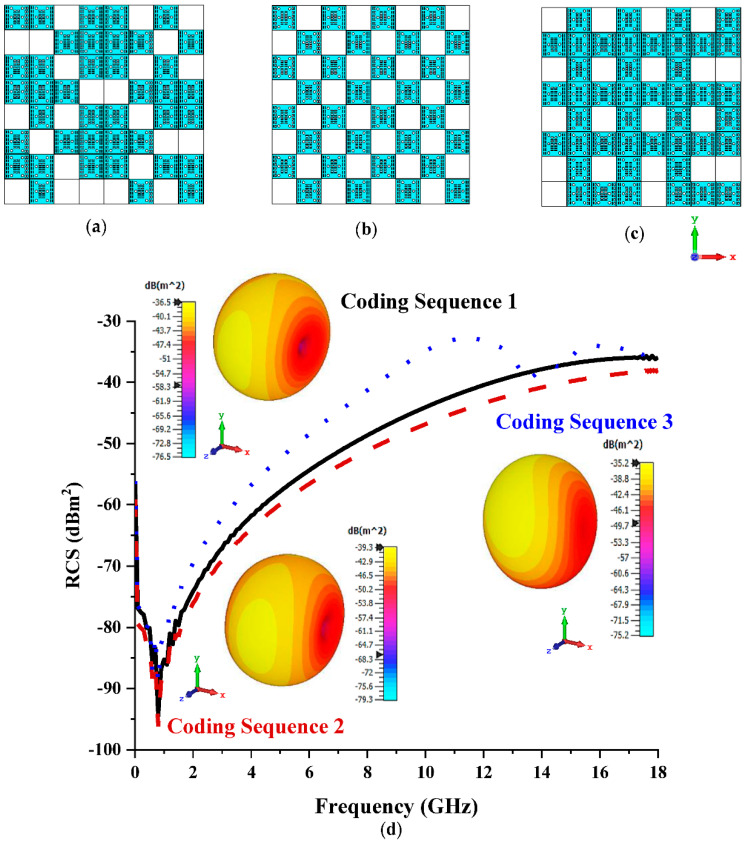
Eight lattices: (**a**) Coding Sequence 1; (**b**) Coding Sequence 2; (**c**) Coding Sequence 3; (**d**) monostatic RCS and bistatic scattering pattern at 15 GHz.

**Figure 4 materials-15-07447-f004:**
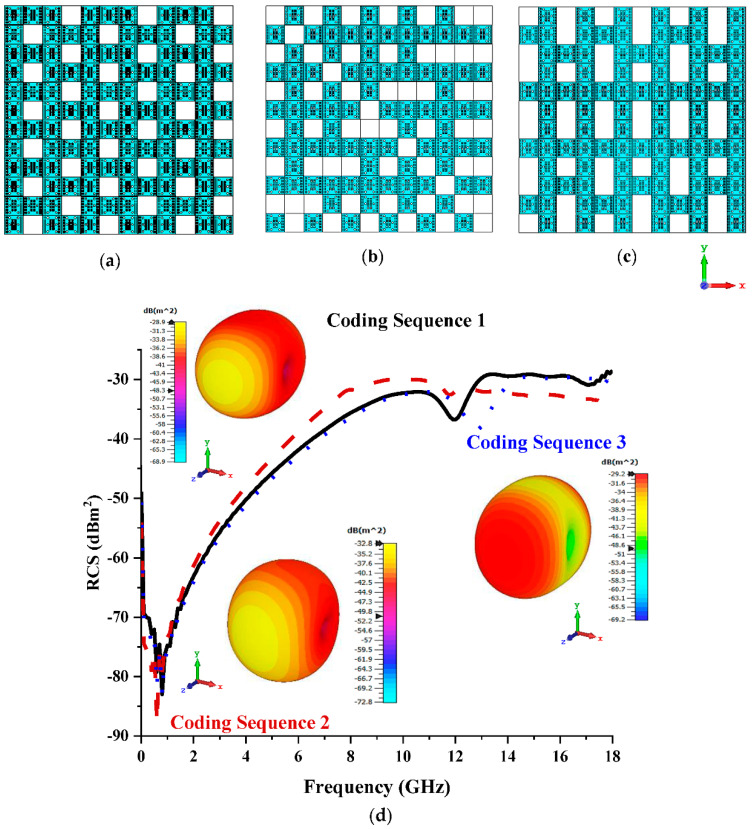
Twelve lattices: (**a**) Coding Sequence 1; (**b**) Coding Sequence 2; (**c**) Coding Sequence 3; (**d**) monostatic RCS and bistatic scattering pattern at 15 GHz.

**Figure 5 materials-15-07447-f005:**
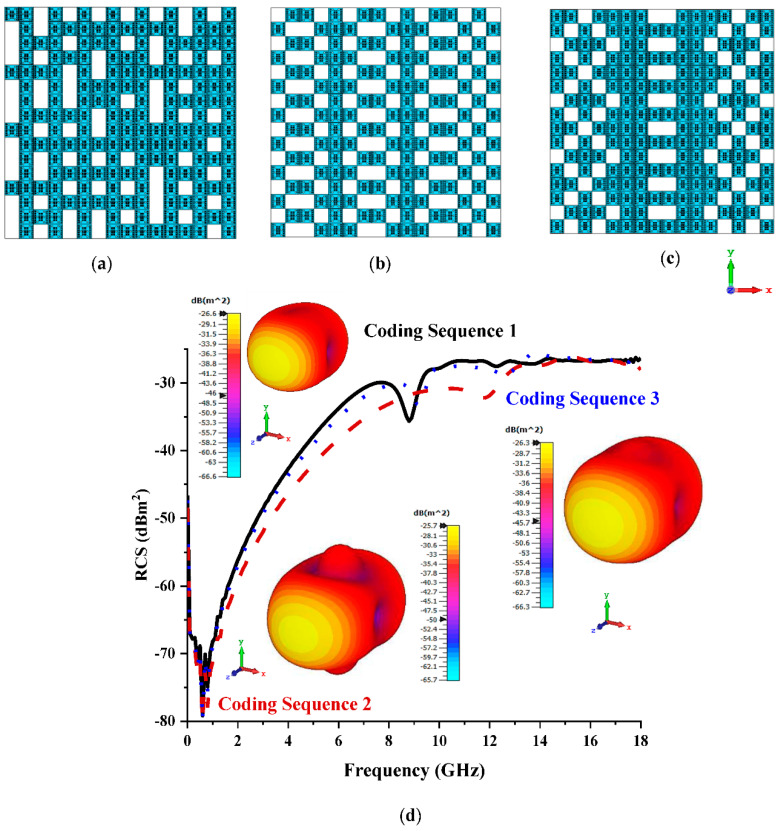
Sixteen lattices: (**a**) Coding Sequence 1; (**b**) Coding Sequence 2; (**c**) Coding Sequence 3; (**d**) monostatic RCS and bistatic scattering pattern at 15 GHz.

**Figure 6 materials-15-07447-f006:**
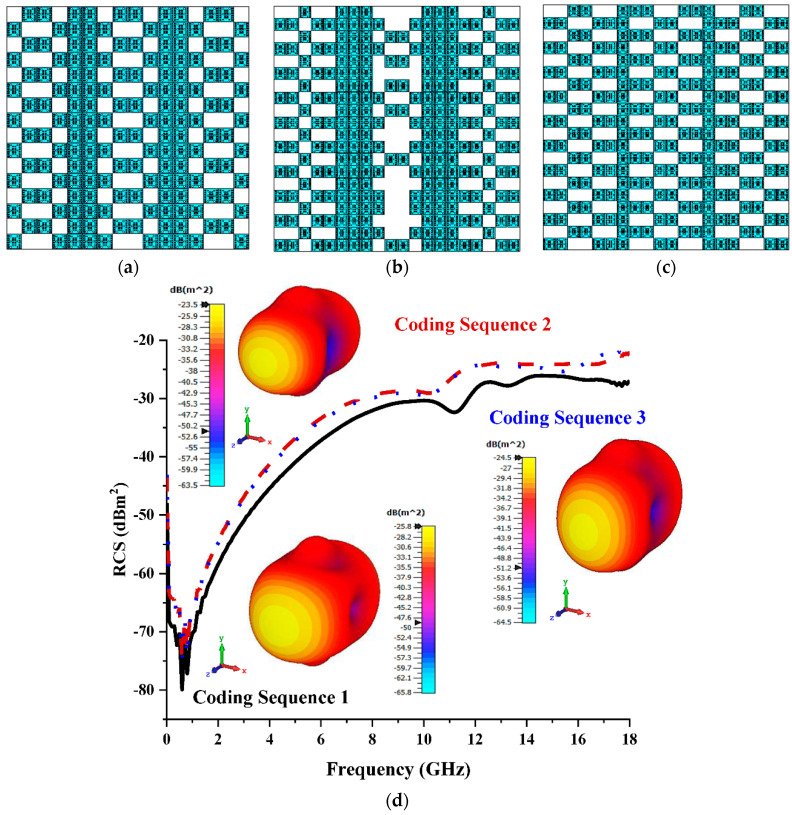
Twenty lattices: (**a**) Coding Sequence 1; (**b**) Coding Sequence 2; (**c**) Coding Sequence 3; (**d**) monostatic RCS and bistatic scattering pattern at 15 GHz.

**Figure 7 materials-15-07447-f007:**
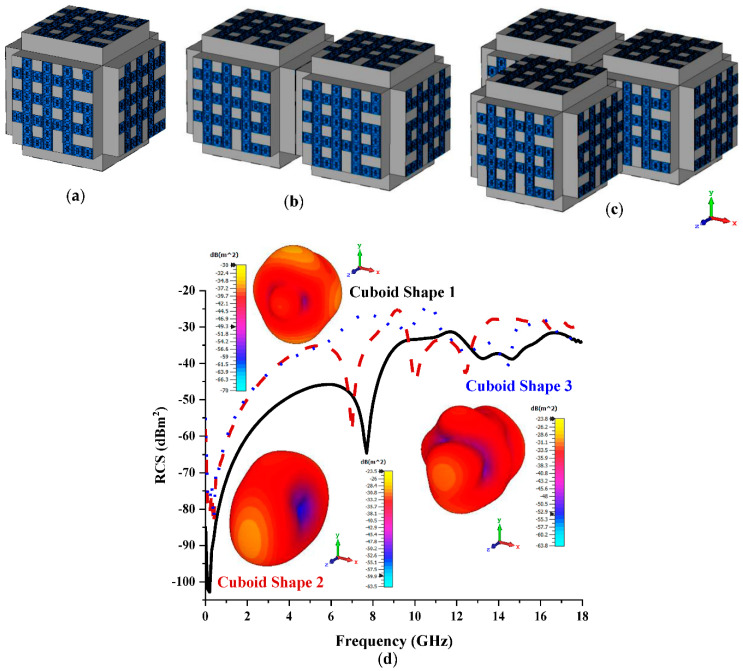
Eight lattices: (**a**) Cuboid Shape 1; (**b**) Cuboid Shape 2; (**c**) Coding Shape 3; (**d**) monostatic RCS and bistatic scattering pattern at 15 GHz.

**Figure 8 materials-15-07447-f008:**
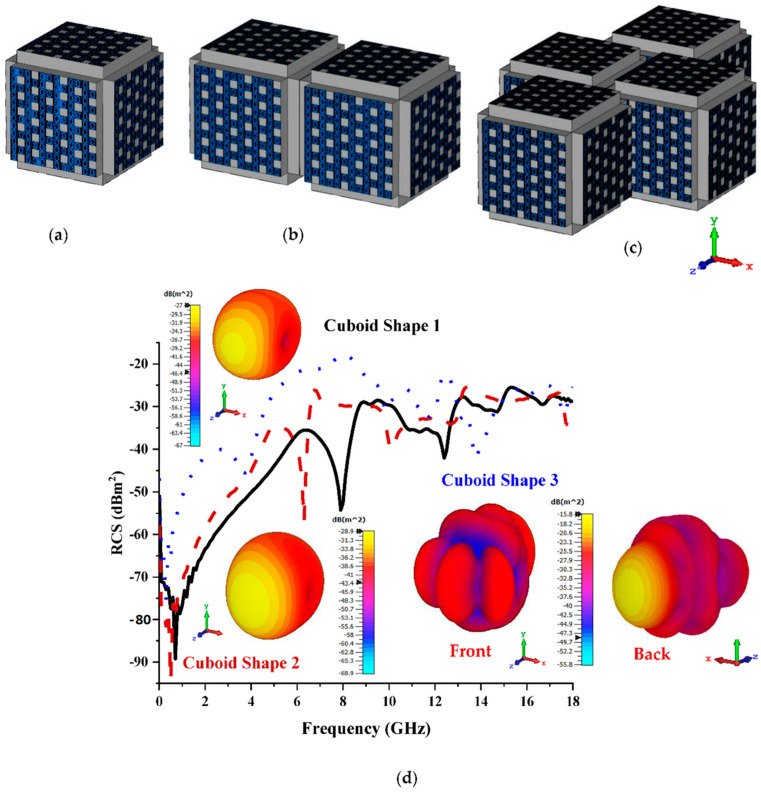
Twelve lattices: (**a**) Cuboid Shape 1; (**b**) Cuboid Shape 2; (**c**) Coding Shape 3; (**d**) monostatic RCS and bistatic scattering pattern at 15 GHz.

**Figure 9 materials-15-07447-f009:**
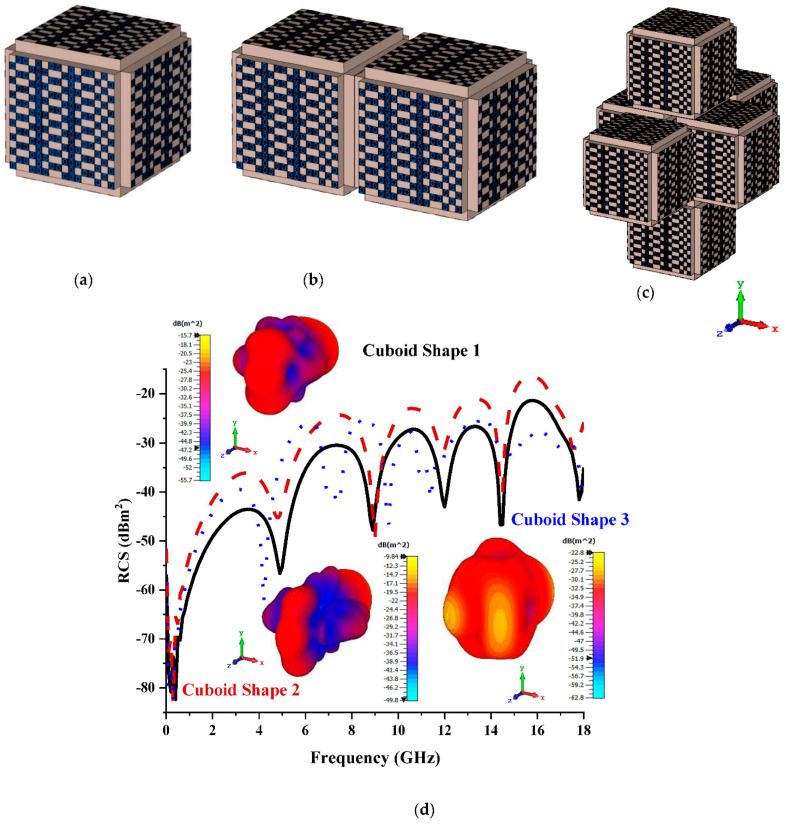
Sixteen lattices: (**a**) Cuboid Shape 1; (**b**) Cuboid Shape 2; (**c**) Coding Shape 3; (**d**) monostatic RCS and bistatic scattering pattern at 15 GHz.

**Figure 10 materials-15-07447-f010:**
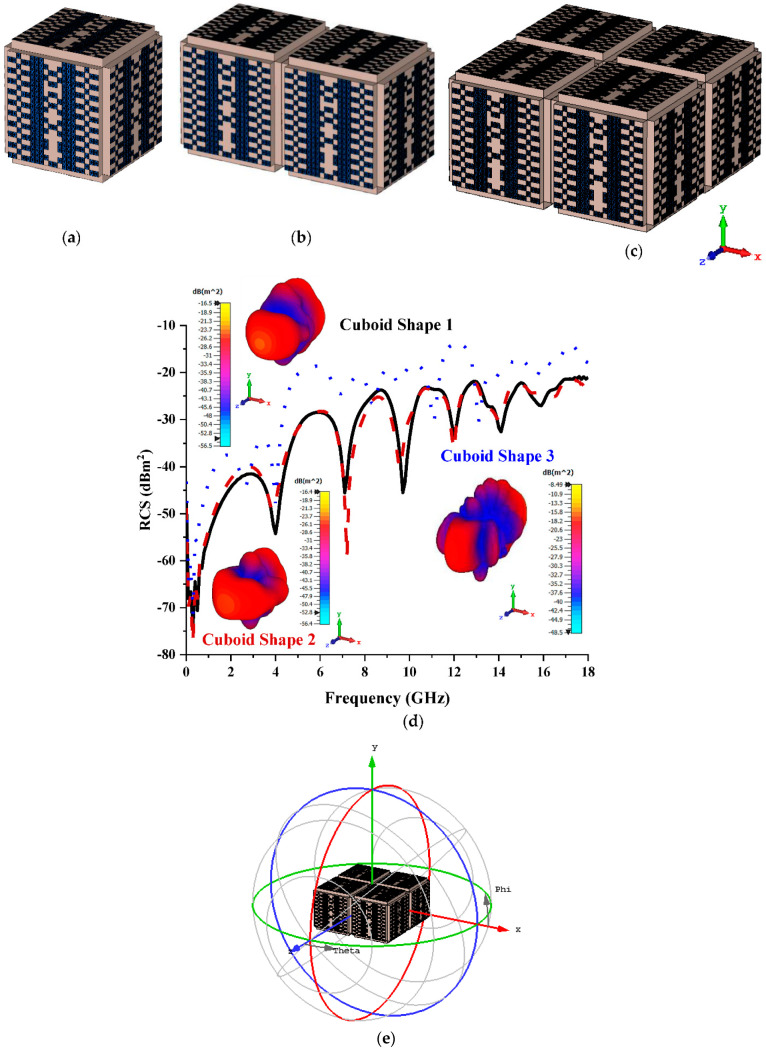
Twenty lattices: (**a**) Cuboid Shape 1; (**b**) Cuboid Shape 2; (**c**) Coding Shape 3; (**d**) monostatic RCS and bistatic scattering pattern at 15 GHz; (**e**) polarisation direction.

**Table 1 materials-15-07447-t001:** Dimensions of element ‘1’.

Description	Dimension (mm)	Description	Dimension (mm)
a	1.0	s4	0.05
b	1.6	t1	0.1
c	0.98	t2	0.03
s1	0.1	t3	0.05
s2	0.024	t4	0.04
s3	0.05	d	0.1

**Table 2 materials-15-07447-t002:** Comparison table of previous literature and proposed design.

References	Dimension of Elements	Best Design Size	Frequency Range (GHz)	Design and Application	Monostatic RCS Reduction (dBm^2^)
[28]	9.5 mm × 9.5 mm	80 mm × 80 mm	2 to 18	Hybrid metamaterial absorber	−30.1 (peak)
[29]	8 mm × 8 mm	48 mm × 48 mm × 192 mm	0 to 18	Coding metamaterial	>−10
[30]	9 mm × 9 mm	80 mm × 80 mm	2 to 12	Polarisation conversion metamaterial	−23 (at 4.3 GHz)
[31]	12.6 mm × 12.6 mm	126 mm × 126 mm	2 to 14	Band-notched absorber	Near 0 (from 8.2 to 9.2 GHz)
Proposed	1 mm × 1 mm	Conventional: 20 mm × 20 mm Cuboid: 40 mm× 40 mm × 20 mm	0 to 18	Coding metamaterial	Conventional: −28 to −23 Cuboid: −31 to −12

## Data Availability

All the data are available within the manuscript.

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
