# Peer review of "Coding Metamaterial Analysis Based on 1-Bit Conventional and Cuboid Design Structures for Microwave Applications"

_materials, 2022, doi:10.3390/ma15217447_

Round 1
Reviewer 1 Report
The manuscript presents investigation of a 1-bit coding metamaterial design for micro-wave application. Various conventional and cuboid shapes assembled by the proposed unit-cells were analyzed. Micro-wave attenuations were investigated under the various configurations.
1. Though the authors claimed 1-bit metamaterial, the tunning mechanism is unclear. All the simulation and results are based upon try-error.
2. What is the design mechanism for the 8 Lattices, 12 Lattices and 16 Lattices?
3. What is the boundary conditions for the simulation? Also, please compare the results with that in previous studies.
Author Response
As attached.

Reviewer 2 Report
The authors present detailed investigations on 1-bit phase-based coding metamaterials. The work is adequate to the scope of the Journal, and can be recommended for publication after the following observations are addressed:
1. The material FR-4 introduced in the abstract must be explained.
2. Why was only the c-parameter modified in the study? What happens to the other parameters?
3. Figure 2a is unreadable at normal zoom, the resolution must be improved. Also, the captions must include what the radiation patterns represent wherever they appear.
4. For the cuboid architecture, the authors must specify the properties of the input radiation field, especially polarization and direction of propagation.
5. The introduction can be supplemented with recent studies on THz metamaterials: https://doi.org/10.3390/polym13111860 and https://doi.org/10.1016/j.jqsrt.2020.107209
I can recommend the manuscript for publication after these observations have been addressed.
Author Response
As attached.

Reviewer 3 Report
For this reviewer, this manuscript presents a valid study for reducing the radar cross-section (RCS) employing a coding metamaterial approach and performing electromagnetic simulations for this objective.
However, the manuscript needs a lot of improvement on its writing.
A more profound analysis of the performed study and its results is necessary to yield more clear conclusions.
A direct application of this study and its results on a real-world device seems necessary to demonstrate how to apply the proposed results.
The authors claim that “this work concentrated on miniaturised unit cell design because earlier literature suggested larger-sized unit cell designs and lattices”. No explicit reference or comparison is made with the suggested “earlier literature”.
This reviewer does not recommend this manuscript to be accepted for publication.
Below are some not exhaustive remarks from which I base my opinion:
An English revision is necessary. Some sentences seem to be directly translated to English, without a proper revision, presenting no significant meaning. This leads to a difficult and hard reading of the manuscript.
Line 274, why was the phase responses “extraordinary”?
Lines 75 and 77, the authors claim that FR-4 is “a suitable material for the construction of the coding metamaterial design.” It is not clear why it is the case.
Figure 2 (a), it is not clear which one of the four cells displayed correspond to each “c” value
Same Figure 2 (a). Why is that for c = 1 mm the phase response is 180 degrees to all frequencies? Why the same happened for c = 0.92 mm, but this time having zero degrees to all frequencies? Was it defined 180° for c=1mm? Those questions are not clearly explained on the manuscript.
Lines 300 and 301, Section 3.2.2 – It is stated that “Coding Sequence 3 yielded the lowest RCS values”, whereas from the graph of Figure 3 (d) and scattering pattern it appears that the Coding Sequence 2 displays the lowest RCS values (on the graph, the line in brown representing Coding Sequence 2 has lower values overall than the line in blue representing Coding Sequence 3). Why is that? A better explanation on the manuscript is needed to why one coding sequence is better than the other. This observation also applies to the other sections that compares different coding sequences.
In which frequency are the displayed patterns of Figures 2 to 10?
Lines 440 to 442, it is stated that “the RCS value can be reduced with larger lattices”. From the showed graphs on Figures 2 to 5, the RCS value seems to increase with a bigger number of cells. A more profound explanation is needed to clarify to eventual readers why this is the case.
Lines 728 to 730, the authors states that “this work concentrated on miniaturised unit cell design because earlier literature suggested larger-sized unit cell designs and lattices”. A reference to the earlier literature and a size comparison is needed
Author Response
As attached.

Reviewer 4 Report
The study aims to investigate the compact 1-bit coding metamaterial design with various conventional and cuboid shapes by analysing the bistatic scattering patterns and as well monostatic Radar Cross-Section for microwave applications. My proposed changes are as follows:
1. Authors are mentioning microwave applications. In this relation it would be highly desirable to compare the obtained results with the experimental outputs.
2. Authors should stress novelty of their work in comparison with others as the topic of the metamaterials studies is the one extensively covered.
3. Authors are mentioning absorption concept. In this relation it would be highly desirable to stress its origin in the context of the manuscript.
4. Authors are missing some recent articles in the field such as Investigation of Hyperbolic Metamaterials.
Author Response
As attached.

Round 2
Reviewer 1 Report
The authors have addressed all the issues.
Author Response
Thank you for the recommendation.
Reviewer 2 Report
The authors have answered my observations. I can recommend for publication.
Author Response
Thank you for the recommendation.
Reviewer 3 Report
This reviewer thanks the authors for their responses. Nevertheless, the opinion of this reviewer is that the present manuscript needs improvement before being considered for acceptance.
Some comments for not recommending acceptance:
Line 791 to 794 – The authors claim that the novelty of this study is defined as the distinct phenomenon of the proposed cuboid-shaped metamaterial exhibit asymmetrical scattering pattern. For this reviewer, from the presented figures it is not clear which cuboid structures present or not this asymmetrical pattern. This should be more evident on the manuscript. A more profound (possible) explanation for this behavior other than only citing the gradient phase distribution would improve the manuscript.
Still on the asymmetrical scattering pattern, how would the authors discard that a certain anomaly on the scattering pattern could not be generated by some numerical electromagnetic simulation error, given that for example the structure from Figure 9 (b) there are at least 1400 “1” elements, each “1” element with 46 geometrical shapes inside (and much more elements for the other structures as the one from Figure 9 (c) or Figure 10 (c))?
Nevertheless, part of the same authors on the referenced published paper [24] already reported this same novel phenomenon (asymmetrical scattering pattern) – and with very similar phrasing as in lines 792 to 795 –, thus this phenomenon cannot be stated as novelty here.
In line with the former question 6 from reviewer 3: Other referenced publications refer to the reduction of the RCS in relation to the same metamaterial structure, but completely metallic, for measuring in dB (and not dBm2) the RCS decrease between the simulated RCSs. However, the authors seem mistakenly to refer to the ‘RCS decrease’ in dBm2 and as the ‘RCS’ itself, as expressed at table 2 and throughout the manuscript. For this reviewer, to numerically and quantitatively measure the RCS decrease caused by the proposed metamaterial structures, a reference structure with an RCS value must be given and comparisons made.
This way, the comparison made at table 2 can make more sense since, for example, the shown value -23 dBm2 for the reference [30], at the publication is actually -23 dB, extracted comparing a metamaterial structure (giving an RCS value in dBm2) and a completely metallic structure with the same dimensions (giving another RCS value in dBm2). And also, it can be clearer to a reader why a cuboid structure having an RCS of -8.49 dBm2 is better than another structure having an RCS of -16.4 dBm2, that is not well explained on the manuscript.
Some additional remarks:
On the answer number 5 for the reviewer 3, the authors provided some insight on how the physical parameters change the phase response of the element.
For this reviewer, to improve the manuscript, some insights (from response 5) should be given on this important part of the manuscript (section 2), such as that “changing the c-value only manifest the desired phase response properties that the authors looking for.” or that “changing the design structure, for example, the size of the circle does not manifest a relative effect on the phase response property.”
Line 406 – it reads “… indicated slightly better RCS reduction values compared to the previous lattices.” For this reviewer, whenever a mention to RCS reduction appears on the manuscript, a numerical comparison on the text is a must for better comprehension.
Line 407, the authors should be explicit on why the behaviors are similar.
Line 455, What “unique properties” are the authors aiming for?
For section 3.2 (3.2.1, 3.2.2, 3.2.3, 3.2.4), it is not clear on the manuscript how one sequence was chosen as the better over the others.
Lines 490 e 491 – How was the 30% reduction calculated? What was the numerical value for the Coding sequence 1? It is not given on the text, and should be given.
Section 3.2.4, line 493, there is no reference for the range of frequencies the shown values refer to.
Line 495 – The size constraint variable mentioned, is it related to the proposed structures to reduce the RCS or is it related to a device in a possible practical application? The manuscript should be clearer.
Figure 8 (d) the scale of the bistatic scattering pattern for cuboid shape 3 is much different from the other shapes, thus the lack of yellow color might induce the reader to a wrong conclusion
Line 775 – This is the first mention on this manuscript to a radiation pattern. Is it “radiation pattern” or “scattering pattern”?
It is not clear on the manuscript the usefulness of the added Figure 10 (e).
For table 2, adding a comparison in terms of lambda (the wavelength) seems appropriate to better compare structures with different sizes working at different frequencies.
The manuscript still needs improvement its writing, some sections are still hard and difficult to understand. Some more comments:
Section 3.1, lines 327 to 329 – it reads “Although the changing of c values manifested unique phase response properties when compared to other parameters and all four designs exhibited similar monostatic RCS values.” For this reviewer, the word “although” in this context seems out of place. A revision of the writing in this important section around this phrase is needed so it is clearer how element 1 was chosen.
Line 340, the division of the word metamaterial should be meta-material, instead of met-amaterial.
Line 344, it is preferable 1 mm x 1 mm, instead of 1 x 1 mm2. This applies to all the other cases.
Line 404 – the first paragraph of section 3.2.2 starts as “Based on the observation, the increased number of lattices affected the RCS reduction values for the 3 different coding sequences.” The text should be clearer on which observation are the authors referring to since this is the begin of the first paragraph of a new section. Also, on the same line, the text is not clear what are numerically the “RCS reduction values”?
Line 414, it reads “that they have to deal with”. It is uncertain form the phrase and context who are “they”.
Line 760 – “On the other”? Something is missing for better understanding.
Author Response
As attached.
